# rGO-WO_3_ Heterostructure: Synthesis, Characterization and Utilization as an Efficient Adsorbent for the Removal of Fluoroquinolone Antibiotic Levofloxacin in an Aqueous Phase

**DOI:** 10.3390/molecules27206956

**Published:** 2022-10-17

**Authors:** Manjot Kaur, Shafali Singh, Surinder Kumar Mehta, Sushil Kumar Kansal

**Affiliations:** 1Department of Chemistry, Centre of Advanced Studies in Chemistry, Panjab University, Chandigarh 160014, India; 2Dr. S. S. Bhatnagar University Institute of Chemical Engineering and Technology, Panjab University, Chandigarh 160014, India

**Keywords:** fluoroquinolones, levofloxacin, heterostructure, adsorption

## Abstract

Herein, the heterostructure rGO-WO_3_ was hydrothermally synthesized and characterized by HRTEM (high-resolution transmission electron microscopy), FESEM (field emission scanning electron microscopy), XRD (X-ray diffraction), FT-IR (Fourier transform infrared spectroscopy), XPS (X-ray photoelectron microscopy), nitrogen physisorption isotherm, Raman, TGA (thermogravimetric analysis) and zeta potential techniques. The HRTEM and FESEM images of the synthesized nanostructure revealed the successful loading of WO_3_ nanorods on the surface of rGO nanosheets. The prepared heterostructure was utilized as an efficient adsorbent for the removal of a third-generation fluoroquinolone antibiotic, i.e., levofloxacin (LVX), from water. The adsorption equilibrium data were appropriately described by a Langmuir isotherm model. The prepared rGO-WO_3_ heterostructure exhibited a Langmuir adsorption capacity of 73.05 mg/g. The kinetics of LVX adsorption followed a pseudo-second-order kinetic model. The adsorption of LVX onto the rGO-WO_3_ heterostructure was spontaneous and exothermic in nature. Electrostatic interactions were found to have played a significant role in the adsorption of LVX onto the rGO-WO_3_ heterostructure. Thus, the prepared rGO-WO_3_ heterostructure is a highly promising material for the removal of emerging contaminants from aqueous solution.

## 1. Introduction

The presence of pharmaceutical substances and their metabolites in the aquatic environment has received increased attention as significant amounts of pharmaceutical compounds have been identified in various water resources, including ground, surface, drinking, tap and sea water [1,2,3,4,5,6]. These substances can reach the environment via industry, wastewater disposal, hospital effluents, anthropogenic effluents and excretion from human beings and animals through urine or faeces [3,4,7,8]. These micro pollutants can enter into wastewater treatment plants (WWTPs) through the sewage system, and most conventional WWTPs are not highly efficient with respect to the complete removal of these persistent contaminants [1,2,3,4,5,9].

Synthetic antibiotics are globally employed in human and veterinary medicines due to their high potency, wide activity spectrum and ease of bioavailability [10,11,12]. Approximately 30–90% of antibiotics are excreted as such or in the form of metabolites through excretion due to their partial metabolism in living beings [13]. Antibiotics possess the property of bioaccumulation, which leads to pathogen resistance, even at low concentrations, and is ultimately accountable for the disruption of the endocrine system [2,3,4,5,6,14,15]. Antibiotics have degradation half-life of 10.6 days in surface water, however in soil matrices, the degradation half-life is 580 days, and are therefore retained in the environment for long durations [16]. Fluoroquinolones (FQs) are non-steroidal man-made antibiotics the exhibit broad activity against Gram (−) and Gram (+) bacteria [17]. First- and second-generation quinolones display remarkable performance against gram (−) bacteria; however, third- and fourth-generation quinolones have extended their substantial activity against both Gram (−) and Gram (+) bacteria [1]. The persistent character of FQs makes them resistive toward hydrolysis, and the partial mineralization of FQs results in the formation of transformation products that still exhibit genotoxic effects [1,10,18,19,20,21,22]. Therefore, the long persistence of FQs in WWTP effluents is responsible for bioaccumulation, genotoxicity and detrimental effects on microbial communities and terrestrial organisms [2,14,18,23]. LVX is a third-generation FQ antibiotic utilized worldwide for the treatment of life-threatening microbial infections [18]. It exhibits resistance toward biodegradation, with approximately 40% removal efficiency achieved in conventional WWTPs [18,24,25]. Therefore, sustainable strategies for the treatment of wastewater should be developed in order to effectively eliminate the persistent emergence of contaminants.

Fluoroquinolones have been removed from wastewater using conventional water purification techniques, such as flocculation, activated sludge, chlorination, ozonation, filtration, precipitation, sedimentation, coagulation, adsorption and reverse osmosis. However, these techniques are subject to some limitations. For example, the coagulation–flocculation technique is only capable of eliminating turbidity and colour and is incapable of removing organic/inorganic contaminants, dissolved impurities and heavy metals [26]. The activated sludge process is subject drawbacks such as sludge expansion, high operating costs, loose floc formation and poor effluent quality [27]. The ozonation technique is commonly used for water disinfection, although it produces carcinogenic bromates as byproducts in the treated water and is also a costly to operate [28]. The filtering process may effectively remove pathogens and turbidity, but it has a poor response to the removal of organic materials and produces an excess of disinfection byproducts when chemical disinfectants are introduced [29]. Membrane technologies, such as reverse osmosis, are limited by the blockage of pores by pollutants, rendering them ineffective in eliminating contaminants within a short period of time. They also emit unpleasant odours as a result of organic, colloidal particle and biological contaminant scaling [28]. In the presence of organic contaminants, the majority of resins become contaminated during the ion-exchange process [30]. The Fenton process generates iron sludge, which causes secondary pollution and is a costly procedure that requires a restricted operating pH range. Additionally, it there are risks associated with reagent storage, handling and transportation [31]. Among all the available techniques, the adsorption process is the most extensively utilized and efficient method for wastewater treatment. The development of cost-effective, environmentally friendly and efficient adsorbents possessing excellent regeneration capabilities is a considerable challenge.

In this regard, the scientific community has been focused on designing high-performance adsorbents, such as MOFs, metal oxides and carbon-based materials, using various approaches [32,33,34,35,36,37,38]. Among these, the coupling of carbonous nanomaterials with semiconductors is a promising approach for the treatment of contaminated water. Among various materials, graphene and its derivatives have attracted increasing interest due to their remarkable electrical conductivity, high mechanical stability, high theoretical specific surface area, exceptional transparency and unique optical characteristics [39,40,41]. Tungsten trioxide (WO_3_), a transition metal-based oxide, possesses intriguing characteristics, such as excellent stability in acidic media, resistance against photocorrosion, non-toxicity and a suitable band gap and is therefore applied in a diverse range of applications, including gas sensing, photocatalysis, electrochemical devices, etc. [42,43,44]. In recent years, WO_3_ has been used as an adsorbent for the removal of ions, such as Sr^2+^ and Cs^+^, due to its ability to provide high stability and excellent ion-exchange capacity [45,46,47]. Therefore, the hybrid nanostructures developed by combining graphene derivatives and WO_3_ might be used for the removal of organic contaminants. This heterojunction provides uniform loading of WO_3_ on the rGO surface, thereby suppressing the aggregation of WO_3_. Moreover, this hybrid nanostructure facilitates the separation of the adsorbent from the aqueous phase, making it a suitable adsorbent material.

In the present work, an rGO-WO_3_ heterostructure was synthesized using the hydrothermal method and characterized using HRTEM, FESEM, XRD, FT-IR, XPS, BET, a surface area analyser, Raman spectroscopy, TGA and zeta potential. The prepared rGO-WO_3_ heterostructure was employed for the adsorptive removal of LVX. Various operational parameters, including pH, adsorbent dose, initial adsorbate concentration and temperature, were optimised in order to achieve maximum antibiotic removal. Moreover, the kinetics and equilibrium adsorption were thoroughly investigated.

## 2. Experimental Section

### 2.1. Materials

All chemicals (analytical grade) were used as provided without any further purification. Sodium tungstate dihydrate (NaWO_4_·2H_2_O, ≥98%), hydrochloric acid (HCl, 36.5–38%) and sodium hydroxide (NaOH, ≥97%) were purchased from Merck, India. Ammonium sulphate ((NH_4_)_2_SO_4_, ≥99%) and oxalic acid (C_2_H_2_O_4_·2H_2_O, 99.8%) were procured from Sigma Aldrich and SD Fine-Chem Ltd., India, respectively. The synthetic antibiotic LVX was provided by Saurav Chemicals Ltd. (Derabassi, India). The solutions were prepared in double-distilled water, and pH was monitored on a Mettler Toledo pH meter (FEP 20) (Greifensee, Switzerland) by adding 0.1 M HCl and NaOH solutions.

### 2.2. Synthesis of rGO-WO_3_ Heterostructure

GO was previously synthesized by our group and utilized as such in this work [11]. For the synthesis of the rGO-WO_3_ heterostructure, 9 mmol of NaWO_4_·2H_2_O was dissolved in 50 mL of water, and the pH of this solution was adjusted to approximately 2 by adding 9 M HCl solution. Then, 27 mmol of C_2_H_2_O_4_·2H_2_O was dissolved in 50 mL of water and added to the above reaction mixture. Furthermore, 5 g of (NH_4_)_2_SO_4_ was added to the resultant suspension and stirred for 1 h. Approximately 148 mg of GO was suspended in 20 mL of water, ultrasonicated for 1 h and added to the reaction mixture. Then, the reaction mixture was placed in an oven for hydrothermal treatment at 175 °C for 27 h. The powder was obtained, repeatedly washed with alcoholic mixture and finally dried. Pure WO_3_ nanorods were synthesized in the same manner, except without the addition of GO.

### 2.3. Characterization

TEM images were captured on an H-7500 (Hitachi, Tokyo, Japan) electron microscope. The HRTEM and selected-area electron diffraction (SAED) pattern was acquired using an FEI Technai F20 microscope (Hillsboro, OR, USA) equipped with an EDAX Inc. elemental analyser. FESEM and elemental mapping images were captured on a Hitachi-8010 microscope (Tokyo, Japan). The XRD patterns were collected using Cu Kα radiation in a 2θ range of 10–80° on a PANalytical X’Pert PRO (Malvern, UK) diffractometer. The FT-IR spectra were obtained on a Spectrum-400 FT-IR/FT-FIR spectrometer (Perkin Elmer, Waltham, MA, USA). An XPS survey was conducted on an Omicron electron spectroscope (Uppsala, Sweden) for chemical analysis with an aluminium anode with 1486.7 eV. The nitrogen physisorption isotherm was collected on a Quantachrome Nova 2000e BET analyser (Boynton Beach, FL, USA). The prepared sample was heated at 150 °C for 6 h before BET analysis. The Raman spectrum was obtained on a Renishaw (in via) microscope (Wotton-under-Edge, UK). TGA thermograms were recorded on an SDT Q600-TA (New Castle, DE, USA) at a heating rate of 20 °C/min in a nitrogen atmosphere. The point of zero charge of the rGO-WO_3_ heterostructure was estimated by a Malvern zeta analyser (Malvern, UK). The UV-vis absorbance spectra were measured on a Systronics-2202 spectrophotometer (Ahmedabad, Gujarat, India).

### 2.4. Adsorptive Removal of LVX Using rGO-WO_3_ Heterostructure

Adsorption experiments were performed in 250 mL Erlenmeyer conical flasks placed in a thermostat shaker (KS 4000i model, IKA) at a shaking rate of 200 rpm and a temperature of 35 °C. The reactions were conducted at pH 4 by with a 100 mL volume of 10 mg/L LVX solution. The desired amount of the prepared rGO-WO_3_ heterostructure was dispersed in LVX solution, and the suspension was shaken for 80 min in order to reach an adsorption–desorption equilibrium. The samples were collected at specific time intervals, and the rGO-WO_3_ heterostructure was separated by filtration through Millipore syringe filters (0.45 µm). The residual concentration of LVX was measured using a UV-vis spectrophotometer. The removal efficiency (%) and adsorption capacity (*q_t_*) were measured using the following equations:(1)$$\mathrm{Removal}\;\mathrm{efficiency}=\frac{C_{0}-C_{t}}{C_{0}}\times 100\%$$
(2)$$\mathrm{Adsorption}\;\mathrm{capacity}\;\left(q_{t}\right)=\frac{(C_{0}-C_{t})V}{m}$$
where *C*_0_ and *C_t_* correspond to initial and residual LVX concentration (mg/L), respectively; *V* is the LVX volume (L); and *m* is the mass of the rGO-WO_3_ heterostructure (g).

The kinetics of LVX adsorption onto the rGO-WO_3_ heterostructure were analysed by a linearized integral form of a pseudo-first-order kinetic model, pseudo-second-order kinetic model and intraparticle diffusion model.

The linearized form of the pseudo-first-order kinetic model can be expressed as:(3)$$ln\left(q_{e}-q_{t}\right)=lnq_{e}-k_{1}t$$
where *q_e_* and *q_t_* are adsorption capacities at equilibrium and a specific time, respectively; *tk*_1_ is the pseudo-first-order rate constant (g mg^−1^ min^−1^); and *t* is the time of the adsorption reaction (min).

The pseudo-second-order kinetic model can be expressed in linearized form as:(4)$$\frac{t}{q_{t}}=\frac{1}{k_{2}q_{e}^{2}}+\frac{t}{q_{e}}$$
where *q_e_* and *q_t_* are adsorption capacities at equilibrium and a specific time, respectively; *k*_2_ is the pseudo-second-order rate constant (g mg^−1^ min^−1^); and *t* is the time of the adsorption reaction (min).

The linear equation for the intraparticle diffusion model can be expressed as:(5)$$q_{t}=k_{ip}t^{1/2}+c_{ip}$$
where *q_t_* is the adsorption capacity at a specific time, *k_ip_* is the diffusion coefficient (mg g^−1^ min^−1/2^), *c_ip_* is the constant of intraparticle diffusion (mg g^−1^) and *t* is the time of the adsorption reaction (min).

The adsorption isotherms were studied by performing reactions with varying concentrations (10 to 50 mg/L) with a fixed rGO-WO_3_ heterostructure dosage (0.45 g/L) and a solution pH of 4. Two- and three-parameter isotherms, i.e., Langmuir, Freundlich, Redlich–Peterson and Jossen isotherms, were employed for the experimental data fitting of equilibrium adsorption.
(6)$$\frac{C_{e}}{q_{e}}=\frac{1}{q_{L}b}+\frac{1}{q_{L}}C_{e}\;(\mathrm{Langmuir}\;\mathrm{model})\;$$
(7)$$log\;q_{e}=log\;K_{F}+\frac{1}{n}log\;C_{e}\;(\mathrm{Freundlich}\;\mathrm{model})\;$$
(8)$$ln\frac{C_{e}}{q_{e}}=\beta lnC_{e}-lnA\;(\mathrm{Redlich}\text{-}\mathrm{Peterson}\;\mathrm{model})\;$$
(9)$$ln\frac{C_{e}}{q_{e}}=-lnH_{J}+F_{J}q_{e}^{p}\;(\mathrm{Jossen}\;\mathrm{model})$$
where *C_e_* is the equilibrium LVX concentration (mg/L), *q_e_* is the equilibrium adsorption capacity (mg/g), *b* is the Langmuir constant (L/mg), *q_L_* is the maximum adsorption capacity, *n* is the Freundlich linearity index, *K_F_* is the Freundlich isotherm constant, *A* is the Redlich–Peterson constant (L g^−1^) and *β* is the Redlich–Peterson model exponent, where 0 ≤ *β* ≤ 1, *H_J_*, *F_J_* and *p* are Jossen isotherm constants; *p* is characteristic of the adsorbent, and *H_J_* and *F_J_* are dependent only on temperature. All the experiments were performed in duplicate.

### 2.5. Application of the Prepared rGO-WO_3_ Heterostructure for Removal of LVX in Real Water Samples

A water sample was collected from the Yamuna River (Yamuna Nagar, Haryana, India) to evaluate the adsorptive removal efficiency of the prepared rGO-WO_3_ heterostructure for removal of LVX. The as-obtained water sample was spiked with 10 mg/L of LVX standard solution and investigated for adsorptive removal of LVX using rGO-WO_3_ adsorbent under optimized experimental conditions.

### 2.6. Stability and Reusability of rGO-WO_3_ Adsorbent

The stability and reusability of the spent rGO-WO_3_ adsorbent were examined for application on a commercial scale. After the first cycle of LVX adsorption, the obtained adsorbent was washed with double-distilled water several times to remove LVX from the surface of the adsorbent. The suspended solution was then centrifuged at a speed of 7000 rpm for 15 min so that the adsorbent settled to the bottom of the centrifuge tube. The supernatant was discarded, and the adsorbent was collected and dried in an oven at 60 °C overnight. The above procedure was repeated four times.

## 3. Results and Discussion

### 3.1. Morphological, Structural, Thermal and Optical Characterizations

The morphology of the rGO-WO_3_ heterostructure was investigated using HRTEM and FESEM techniques. TEM images of pure WO_3_ exhibited rod-shaped morphology with length and width dimensions in the nanometre range (Figure 1a,b). HRTEM micrographs of the rGO-WO_3_ heterostructure demonstrated the dense growth of WO_3_ nanorods onto the rGO surface, as shown in Figure 2a–d. The fringe spacing of 0.32 nm matched well with the (102) crystallographic plane of WO_3_ nanorods (Figure 2e). The SAED pattern showed the formation of well-defined spots, indicating the presence of multiple lattice planes and, thereby establishing the polycrystalline nature of the rGO-WO_3_ heterostructure (Figure 2f).

FESEM images of the synthesized rGO-WO_3_ heterostructure are displayed in Figure 3a–e. The FESEM micrographs depict the loading of WO_3_ nanorods on the surface of wrinkled rGO nanosheets, efficiently preventing the restacking of rGO nanosheets. The HRTEM and FESEM images exhibited consistency with each other in terms of dimensionality and morphology. Furthermore, these micrographs confirmed the strong interfacial contact between rGO nanosheets and WO_3_ nanorods. The elemental mapping images showed the distribution of W, O and C in the prepared heterostructure (Figure 3f–i).

The structural phase and purity of the synthesized WO_3_ nanorods and rGO-WO_3_ heterostructure were examined using the XRD technique (Figure 4a). The XRD pattern of WO_3_ nanorods exhibited well-defined diffraction reflections at 2θ = 13.9°, 23.4°, 24.2°, 27.5°, 28.0°, 33.9°, 36.8°, 42.6°, 44.5°, 47.9°, 49.5°, 51.8°, 55.5°, 56.4°, 57.8°, 63.3°, 71.5° and 78.6°, corresponding to the presence of (100), (002), (110), (102), (200), (112), (202), (300), (212), (004), (220), (310), (222), (204), (312), (402), (224) and (413) crystallographic planes, respectively. The XRD data of the WO_3_ nanorods matched well with the standard JCPDS card 01-085-2459 and revealed the presence of a hexagonal phase. The XRD diffractogram of the rGO-WO_3_ heterostructure displayed a similar diffraction pattern with respect to WO_3_ nanorods; therefore, the crystal structure of the WO_3_ nanorods is not affected by the addition of rGO. No apparent peak corresponding to GO was found due to its limited content, weak diffraction intensity and poor crystallinity, as GO was reduced to rGO during the hydrothermal treatment. Crystallite sizes of 61.20 nm and 24.47 nm were measured using Scherrer’s formula for pure WO_3_ nanorods and the rGO-WO_3_ heterostructure, respectively.

The FTIR spectra of pure WO_3_ nanorods and the rGO-WO_3_ heterostructure are shown in Figure 4b. FTIR bands for pure WO_3_ nanorods were found at 805, 1405, 1600, 2908 and 3456 cm^−1^. The peak centred at 805 cm^−1^ can be attributed to the stretching vibrations of W-O-W bonding [43]. The vibrational band at 1405 cm^−1^ can be attributed to the C-OH functional moiety [43,48]. A peak at 2940 cm^−1^ corresponded to the presence of C-H bonding [49]. The peaks at 1600 and 3456 cm^−1^ were assigned to the –OH bending and stretching vibrations of water molecules [42,44,48,50]. The FT-IR spectrum of the rGO-WO_3_ heterostructure exhibited similar FT-IR peaks with respect to pure WO_3_ nanorods.

The XPS analysis provides information regarding valence states, surface composition and the molecular structure of the synthesized rGO-WO_3_ heterostructure. The full XPS survey of rGO-WO_3_ heterostructure showed the presence of C, O and W without any impurities (Figure 5a). The XPS spectrum of C1s displayed peaks at 284.4 and 285.6 eV assigned to graphitic sp^2^ carbon (C=C) and C-O, respectively [43,44,51,52] (Figure 5b). The XPS scan of W showed well-documented peaks at 37.86 and 35.73 eV (Figure 5c). The peaks at 37.8 and 35.7 eV indicated the presence of W4f_5/2_ and W4f_7/2_, respectively [42,44,53]. These findings show that W is present in the W^6+^ oxidation state in the prepared heterostructure. The deconvoluted XPS spectrum of O can be well-resolved into three components, i.e., 533.1, 531.7 and 530.6 eV (Figure 5d). The peak located at 533.19 eV might be related to the existence of a C-OH group [54]. The binding energy peak at 531.79 eV is accredited to a surface-adsorbed O-H group. The binding energy peak at 530.69 eV corresponded to the presence of W-O bonding [43,44,52].

The textural characteristics and porosity of the synthesized rGO-WO_3_ heterostructure were examined using nitrogen adsorption and desorption measurements. The nitrogen physisorption isotherm showed that the prepared heterostructure possessed a specific surface area and total pore volume of 48.823 m^2^/g and 1.582 × 10^−2^ cm^3^/g, respectively (Figure 6a). The pore diameter was determined using the density functional theory (DFT) method (approximately 2.86 nm) (Figure 6b). The larger surface area of the synthesized rGO-WO_3_ heterostructure allowed for good interactions between the heterostructure and the target organic contaminant by offering more reactive sites. In this way, rGO behaved as an excellent supporting matrix for the immobilization of WO_3_ nanorods. Thus, it reduced the agglomeration of WO_3_ nanorods and further enhanced the performance of the synthesized heterostructure.

Raman spectroscopy is a useful microstructure analysis technique for the investigation of structural and electronic properties of nanomaterials. The Raman spectrum of the rGO-WO_3_ heterostructure showed peaks at 274, 773, 917, 1348 and 1593 cm^−1^ (Figure 7). The peaks at 274, 773 and 917 cm^−1^ were related to the presence of WO_3_ nanorods. D and G bands were observed at 1348 and 1593 cm^−1^, respectively [51,52,55,56]. The D band established the presence of a defective carbon structure, including edges and imperfections of sp^3^-hybridized carbon, and the G band is associated with the existence of well-defined sp^2^-bonded carbon atoms. The peak located at 917 cm^−1^ is attributed to the symmetric stretching vibrations of W-O bonds [42,47,56]. The peak positioned at 773 cm^−1^ corresponded to the stretching vibrational modes of O-W-O units [54,57]. The peak at 274 cm^−1^ is related to the bending mode of a W-O-W group [51,52,54,55]. The Raman spectrum of the rGO-WO_3_ heterostructure showed consistency with other characterization techniques, verifying the successful formation of the rGO-WO_3_ heterostructure.

The stability of the prepared WO_3_ nanorods and rGO-WO_3_ heterostructure with respect to temperature was examined using TGA. The TGA thermogram of pure WO_3_ nanorods displayed a total weight loss of 15.34% (Figure 8a). The progressive weight loss of 12.79% up to 500 °C was due to the loss of physically adsorbed water and pyrolysis of oxygen-containing functional groups. The weight loss after 500 °C may be related to the pyrolysis of the carbon skeleton. However, the synthesized rGO-WO_3_ heterostructure exhibited a weight reduction of 4.37% and 6.98% up to 500 °C and 1000 °C, respectively (Figure 8b). Therefore, the rGO-WO_3_ heterostructure exhibited a more enhanced thermal stability than pure WO_3_ nanorods due to the stronger interactions between WO_3_ nanorods and rGO nanosheets and low oxygen content in the synthesized heterostructure. The HRTEM and FESEM characterizations exhibited the dense dispersion of WO_3_ nanorods onto rGO nanosheets, thereby confirming the presence of strong interactions between both components. Moreover, the enhanced thermal stability of rGO is reported in the literature due to the loss of some functional moieties; therefore, the prepared rGO-WO_3_ heterostructure possessed excellent thermally stability [11,58,59].

### 3.2. Utilization of the Synthesized rGO-WO_3_ Heterostructure as an Adsorbent for the Removal of LVX from Water

The synthesized rGO-WO_3_ heterostructure was applied as an efficient adsorbent for the removal of antibiotic LVX. The effect of pH on the adsorptive removal of LVX was determined by varying the pH from 2 to 8 at a fixed adsorbent dose (0.25 g/L) and a fixed temperature (35 °C).

Removal efficiencies of 82.1%, 89.8%, 88.3% and 54.7% were achieved for LVX using the rGO-WO_3_ heterostructure at pH 2, pH 4, pH 6 and pH 8, respectively (Figure 9a). The maximum LVX removal efficiency was achieved at pH 4. A graph showing adsorption capacities vs. time at varying pH values is shown in Figure 9b. The adsorption capacity was increased from 29.15 mg/g to 30.96 mg/g with an increase in solution pH from 2 to 4. The adsorption capacity was then decreased to 30.01 mg/g with increased in pH up to 6. The adsorption capacity was then reduced to 20.01 mg/g at pH 8; therefore, further batch sorption experiments were performed at the optimal pH value of 4.

The zeta potential was recorded for the rGO-WO_3_ heterostructure at varying pH values in the range of 2 to 8 (Figure 10). The synthesized heterostructure is negatively charged throughout the whole pH range. LVX is zwitterionic in the pH region of 5.9–7.9, positively charged at pH values less than 5.9 and negatively charged above pH 7.9. The heterostructure possessed maximum negative charge at pH 4. The maximum adsorption capacity was measured at pH 4 due to the strong electrostatic attraction between the negatively charged rGO-WO_3_ heterostructure and the positively charged LVX. At pH 2, the heterostructure had sufficient negative charge to attract cationic LVX and therefore possessed high sorption capacity. At pH 6, electrostatic interactions between the zwitterionic LVX solution and the negatively charged heterostructure might play an imperative role in the excellent adsorption of LVX onto the prepared heterostructure. The adsorption capacity was minimal at pH 8 due to the enhanced electrostatic repulsion between the negatively charged heterostructure and the anionic LVX solution.

The effect of rGO-WO_3_ heterostructure loading on the removal of LVX was investigated by introducing varying amounts of heterostructure to 100 mL of LVX solution (10 mg/L). Removal efficiencies of 89.84%, 96.16%, 99.27% and 98.66% were obtained with 0.25 g/L, 0.35 g/L, 0.45 g/L and 0.55 g/L heterostructure, respectively (Figure 11a). In this way, the removal of LVX was increased up to 0.45 g/L and further increment in rGO-WO_3_ heterostructure loading caused a slight change in the removal efficiency. The maximum LVX removal was achieved at 0.45 g/L of rGO-WO_3_ heterostructure; therefore, this dosage was selected for subsequent experiments. Adsorption capacities of 30.01 mg/g, 26.56 mg/g, 18.24 mg/g and 15.34 mg/g were achieved with 0.25 g/L, 0.35 g/L, 0.45 g/L and 0.55 g/L of rGO-WO_3_ heterostructure, respectively (Figure 11b).

The impact of initial LVX concentration on the sorption capacity was explored by varying the concentration from 10 to 50 mg/L at pH 4 and adsorbent amount of 0.45 g/L. The adsorption capacity was significantly enhanced from 18.24 mg/g to 70.47 mg/g with increasing LVX concentration from 10 to 50 mg/L (Appendix A), which can be attributed to the increased concentration gradient enabling rapid transfer of LVX molecules onto the surface of the rGO-WO_3_ heterostructure [60]. The kinetics of the adsorption process were studied using a pseudo-first-order kinetic model, pseudo-second-order kinetic model and intraparticle diffusion model in order to obtain insights from the mechanistic point of view (Appendix A). The regression coefficients and kinetic parameters for all kinetic models are listed in Table 1, demonstrating that the adsorption data are well-fitted with the pseudo-second-order kinetic model. Therefore, adsorption of LVX onto the rGO-WO_3_ heterostructure surface is more appropriately described by a pseudo-second-order model. Thus, the adsorption capacity value calculated from the fitting of the pseudo-second-order model matched accurately with the experimental value, verifying that adsorption of LVX over rGO-WO_3_ is governed by chemisorption [61,62].

The reaction temperature strongly affected the adsorption; its impact was observed by varying the temperature from 30 °C to 40 °C (Appendix A). An adsorption capacity of 20.86 mg/g was achieved at 30 °C and reduced to 18.32 mg/g when the temperature was increased to 35 °C. The adsorption capacity was further reduced to 18.03 mg/g with an increase in temperature to 40 °C. The decrease in adsorption capacity with increased temperature confirmed the exothermic nature of the adsorption phenomena [63]. Furthermore, the thermodynamic parameters were determined by plotting a graph of ln *K_c_* vs. 1/T (Appendix A). The Gibbs free energy (Δ*G^o^*) was calculated using the following equation:$$\Delta G^{o}=-RTlnK_{c}$$
where *R* is the gas constant (8.314 J/mol/K), *K_c_* is the adsorption equilibrium constant and *T* is the reaction temperature in Kelvin.

The van’t Hoff equation was used to measure the enthalpy (Δ*H^o^*) and entropy (Δ*S^o^*); its mathematical form is:$$lnK_{c}=\frac{\Delta S^{o}}{R}-\frac{\Delta H^{o}}{RT}$$

The calculated thermodynamic parameters are given in Table 2. The Δ*G^o^* values were measured as negative values for different temperatures, suggesting the feasible and spontaneous nature of the adsorption process. The negative value of Δ*H^o^* inferred that the LVX adsorption onto the rGO-WO_3_ heterostructure is exothermic in nature, and the negative value of Δ*S^o^* indicated the decreased randomness at the solution–solid interface [63,64,65].

Langmuir, Freundlich, Redlich-Peterson and Jossen isotherms were used to fit the equilibrium adsorption data and obtain insights into adsorbate–adsorbent interactions (Appendix A). The isotherm parameters calculated from based on these models are shown in Table 3. The Langmuir model provided a better fit than all other models, indicating that the monolayer adsorption of LVX on the surface of the rGO-WO_3_ heterostructure occurred with a finite number of adsorption sites and uniform energies. An adsorption capacity of 73.05 mg/g was obtained with the Langmuir model.

The adsorption results are compared with the reported literature and listed in tabular form in Table 4, demonstrating that the synthesized heterostructure displayed excellent adsorption capacity toward the adsorption of LVX.

### 3.3. Application of the Prepared rGO-WO_3_ Heterostructure for Removal of LVX in Real Water Samples

The feasibility of the prepared rGO-WO_3_ heterostructure for adsorptive removal of LVX was investigated in a real river water system and compared with a double-distilled water system. As shown in Appendix A, a considerable reduction in adsorptive removal efficiency (55.98%) of adsorbent was observed in the real water system as compared to that of the double-distilled water system (99.27%). The significant decrease in the real water system might be due to the existence of other pollutants (organic, inorganic and natural matter) in real wastewater, which hindered the effective adsorption of LVX on the adsorbent’s surface [75,76]. The obtained results imply that the prepared rGO-WO_3_ adsorbent can be utilized for the removal of LVX in real water samples.

### 3.4. Stability and Reusability Studies

To demonstrate the feasibility of applying the prepared rGO-WO_3_ adsorbent for removal of LVX on a commercial scale, the reusability and regeneration ability of the adsorbent were investigated for four cycles of adsorption under optimized experimental conditions. Figure 12 illustrates the removal efficiencies of rGO-WO_3_ adsorbent for four consecutive cycles of adsorption. The removal efficiency decreased to 70.35% after the fourth regeneration cycle, possibly due to the loss of adsorbent during recovery in each cycle. However, after four cycles, the removal efficiency was still 70% of that achieved in the initial cycle, implying stability with respect to the removal of LVX using rGO-WO_3_ adsorbent.

The FESEM images of the synthesized heterostructure after four consecutive cycles of adsorption are shown in Appendix A. The FESEM images displayed some breakage of the nanorods, although they still adhered to the surface of the rGO sheets. This heterojunction between WO_3_ nanorods and rGO sheets aids in the separation of the adsorbent from aqueous solution.

## 4. Conclusions

In summary, an rGO-WO_3_ heterostructure was synthesized using a hydrothermal approach and displayed intriguing morphological, structural and thermal features. The synthesized heterostructure possessed a surface area of 48.823 m^2^/g and displayed increased thermal stability with respect to pure WO_3_ nanorods. The prepared rGO-WO_3_ heterostructure was utilized for the removal of LVX from synthetic and real wastewater. The adsorption of LVX onto the rGO-WO_3_ heterostructure followed a pseudo-second-order kinetic model. The adsorption of LVX was found to be satisfactorily fit in accordance with Langmuir isotherm model and exhibited an adsorption capacity of 73.05 mg/g. Furthermore, the synthesized rGO-WO_3_ heterostructure displayed good stability for the removal of LVX for up to four cycles of adsorption, implying its utility at a large scale. Therefore, the present work highlights the significance of graphene-derivative-based heterostructures in the field of environmental remediation.

## Figures and Tables

**Figure 1 molecules-27-06956-f001:**
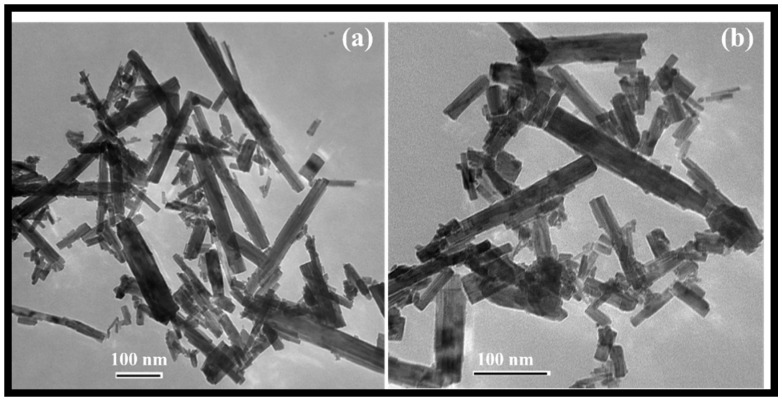
Typical (**a**,**b**) TEM images of WO_3_ nanorods.

**Figure 2 molecules-27-06956-f002:**
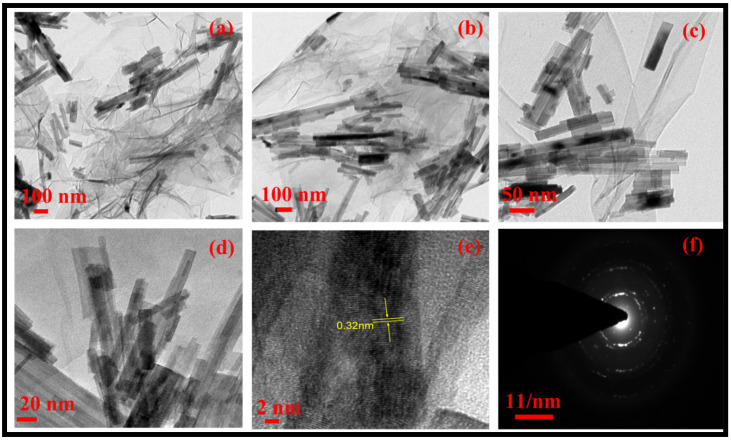
Typical (**a**–**e**) HRTEM images and (**f**) SAED pattern of the rGO-WO_3_ heterostructure.

**Figure 3 molecules-27-06956-f003:**
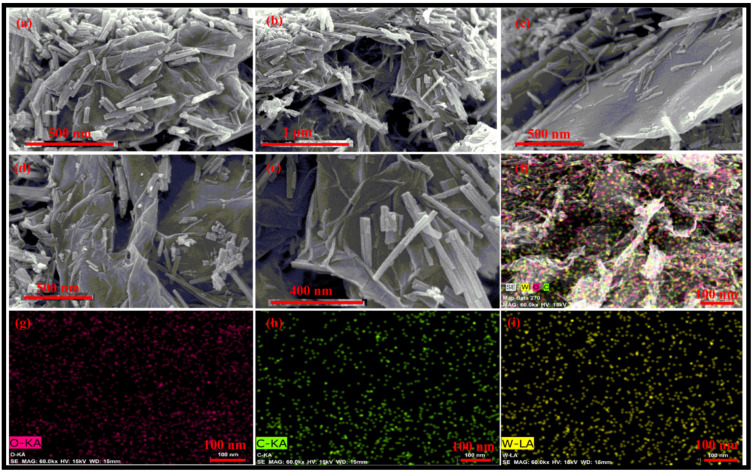
Typical (**a**–**e**) FESEM micrographs of the prepared rGO-WO_3_ heterostructure and elemental mapping images of (**f**) the rGO-WO_3_ heterostructure, (**g**) W, (**h**) O and (**i**) C elements.

**Figure 4 molecules-27-06956-f004:**
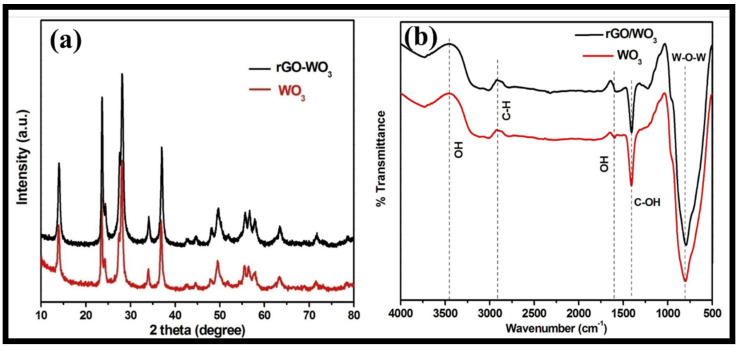
Typical (**a**) XRD and (**b**) FTIR spectra of WO_3_ nanorods and the rGO-WO_3_ heterostructure.

**Figure 5 molecules-27-06956-f005:**
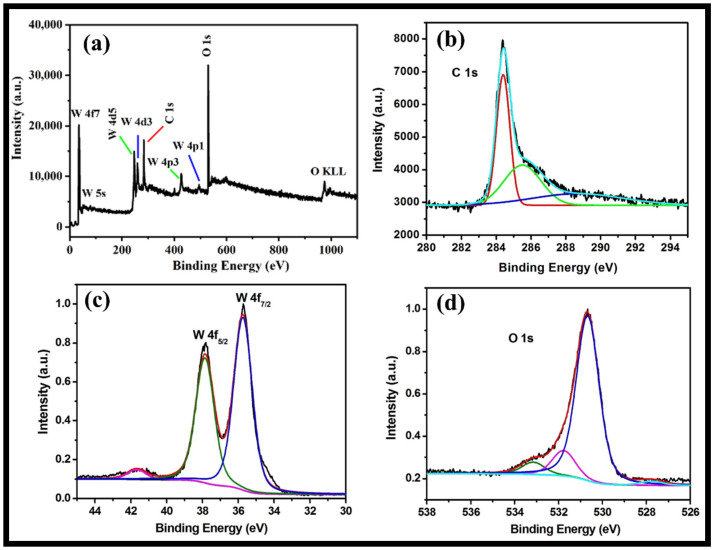
Typical high-resolution XPS survey of (**a**) the rGO-WO_3_ heterostructure and (**b**) C, (**c**) W and (**d**) O elements.

**Figure 6 molecules-27-06956-f006:**
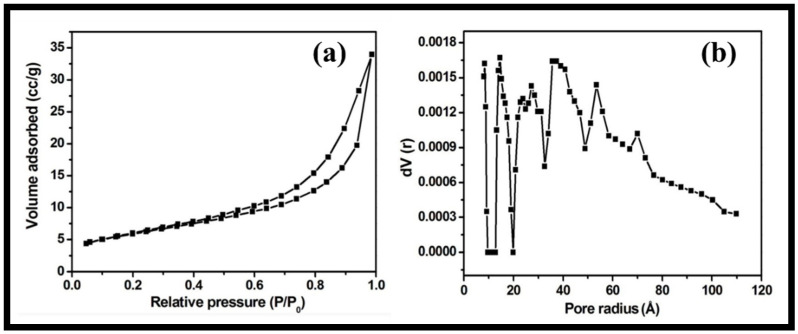
Typical (**a**) nitrogen adsorption–desorption isotherm and (**b**) DFT pore size distribution of the synthesized rGO-WO_3_ heterostructure.

**Figure 7 molecules-27-06956-f007:**
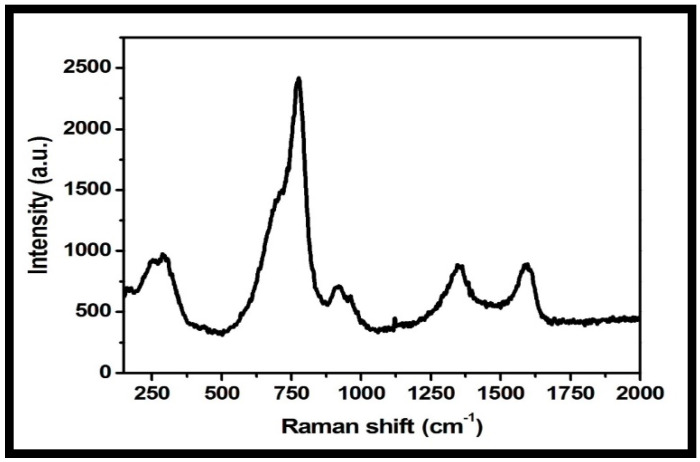
Typical Raman spectrum of the prepared rGO-WO_3_ heterostructure.

**Figure 8 molecules-27-06956-f008:**
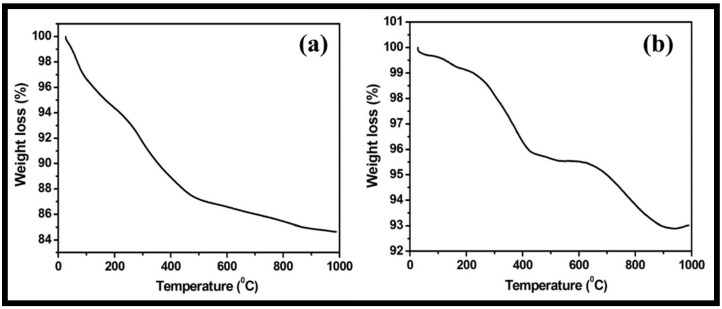
Typical TGA thermograms of (**a**) WO_3_ nanorods and (**b**) rGO-WO_3_ heterostructure.

**Figure 9 molecules-27-06956-f009:**
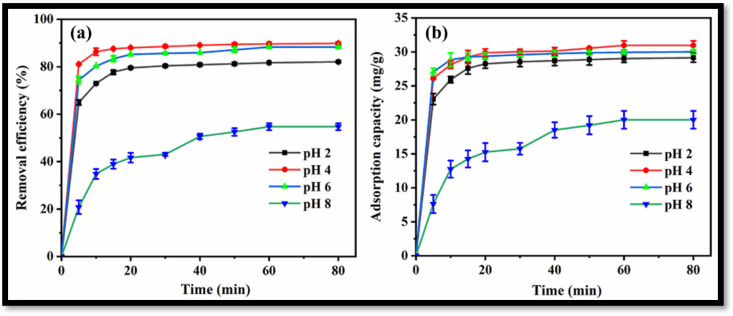
(**a**) Removal efficiency and (**b**) adsorption capacity of rGO-WO_3_ heterostructure at varying pH values.

**Figure 10 molecules-27-06956-f010:**
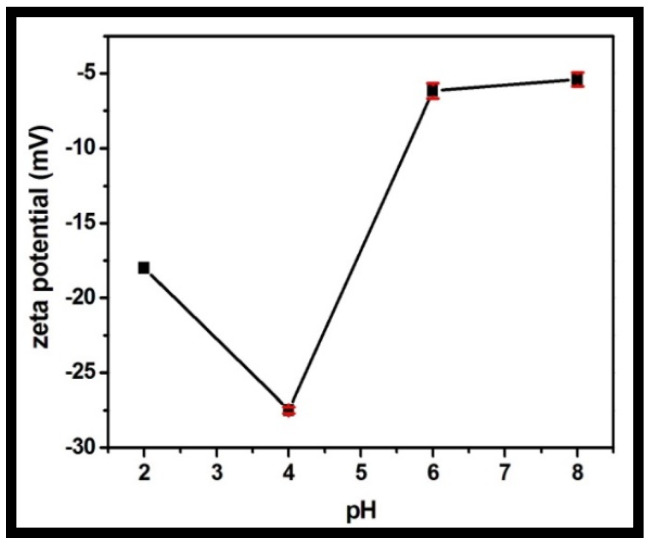
Zeta potential of the prepared rGO-WO_3_ heterostructure with respect to pH.

**Figure 11 molecules-27-06956-f011:**
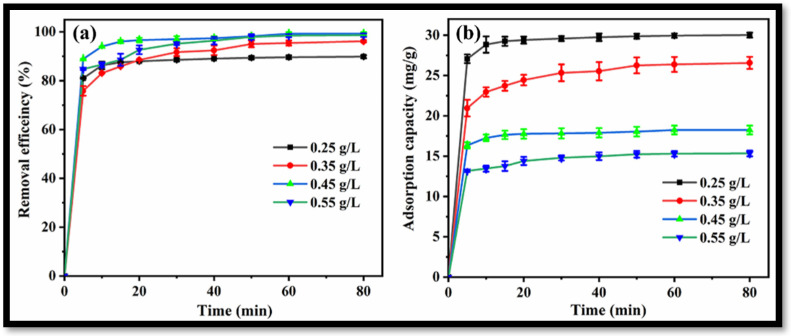
Effect of rGO-WO_3_ heterostructure dosage on (**a**) removal efficiency and (**b**) adsorption capacity.

**Figure 12 molecules-27-06956-f012:**
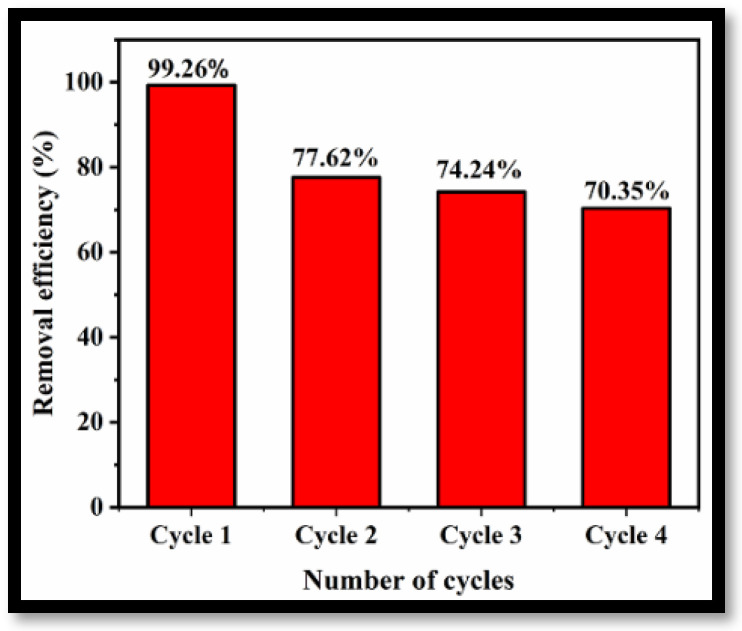
Reusability of rGO-WO_3_ adsorbent for removal of LVX.

**Table 1 molecules-27-06956-t001:** Kinetic parameters of pseudo-first-order, pseudo-second-order and intraparticle diffusion models for the adsorption of LVX using the rGO-WO_3_ heterostructure.

Kinetic Parameters	Concentration				
Pseudo-First-Order	10 mg/L	20 mg/L	30 mg/L	40 mg/L	50 mg/L
** *R* ** ** ^2^ **	0.621	0.733	0.529	0.779	0.771
** *k* ** ** _1_ **	−0.067	−0.143	−0.091	−0.081	−0.079
** *q_e_* ** ** _(exp)_ ** **(mg/g)**	3.25	8.66	6.59	16.04	19.43
** *q_e_* ** ** _(cal)_ ** **(mg/g)**	18.24	31.88	45.79	59.57	70.47
**Pseudo-second-order**					
** *R* ** ** ^2^ **	0.999	0.999	0.999	0.999	0.999
** *k* ** ** _2_ **	0.101	0.079	0.069	0.018	0.018
** *q_e_* ** ** _(exp)_ ** **(mg/g)**	18.18	31.25	45.45	58.82	71.43
** *q_e_* ** ** _(cal)_ ** **(mg/g)**	18.24	31.88	45.79	59.57	70.47
**Intraparticle diffusion model**					
** *R* ** ** ^2^ **	0.425	0.393	0.370	0.454	0.446
** *k* ** ** _3_ **	1.442	2.456	3.461	4.728	5.622
** *c_ip_* **	8.813	15.917	23.389	27.467	33.160

**Table 2 molecules-27-06956-t002:** Thermodynamic parameters for the adsorption of LVX onto the rGO-WO_3_ heterostructure.

Δ*G* (kJ/mol) (303 K)	Δ*G* (kJ/mol) (308) K	Δ*G* (kJ/mol) (313 K)	Δ*H* (kJ/mol)	Δ*S* (kJ/mol.K)
−17.05743	−15.18116	−13.5691	−110.975355	−310.823546

**Table 3 molecules-27-06956-t003:** Equilibrium adsorption isotherm parameters obtained for LVX adsorption using **rGO**-WO_3_ heterostructure.

Langmuir	Freundlich	Redlich-Peterson	Jossen
*R*^2^ = 0.99033	*R*^2^ = 0.96044	*R*^2^ = 0.98928	*R*^2^ = 0.93325
*q_L_* (mg/g) = 73.05	*K_F_* (mg/g) = 36.234	*β* = 0.65575	*F_J_* = 0.03286
*b* (L/mg) = 1.68	*n* = 4.066	*A* = 23.449	*H_J_* = 51.125

**Table 4 molecules-27-06956-t004:** Comparison of the present work with the reported literature for the adsorption of LVX.

Adsorbent	Adsorption Conditions	Specific Surface Area (m^2^/g)	Concentration (mg/L)	Adsorption Capacity (mg/g)	Reference
Natural zeolite	0.5 g/L, 25 °C, pH 6.5, 180 rpm, 2 h	-	5–50	22.17	[66]
Magnetic nanoparticles	1 g/L, 240 min, pH 6.5	-	2.5–20	6.848	[67]
ZIF-8-derived hollow carbon	5 mg, pH 7, 1000 rpm	807.56	5–40	227.8	[68]
WCM 315	2 g/L 24 h, 125 rpm	2.70	15–150	7.38	[69]
WCM 330	2 g/L 24 h, 125 rpm	2.65	15–150	20.4	[69]
WCM 615	2 g/L 24 h, 125 rpm	266	15–150	14.8	[69]
WCM 630	2 g/L 24 h, 125 rpm	225	25–200	25.2	[69]
Iron-pillared montmorillonite	0.5 g/L, 48 h, 45 °C, pH 7	127	20–100	56.66	[70]
Wood chip biochar	10 g/L, 24 h, 30 °C, pH 6.5	312	30–200	7.72	[71]
Rice husk biochar	10 g/L, 24 h, 30 °C, pH 8.0	168	15–150	4.99	[71]
UiO-66/CA	0.32 g wet beads, pH 7, room temperature 150 rpm	549.33	10–1000	86.43	[72]
Zr-modified corn bracts	2 g/L, pH 7, 240 rpm	2.953	100–550	73.10	[73]
WLC	1 g/L, 24 h, 30 °C, pH 6.5, 125 rpm	-	15–150	14.2	[61]
WHC	1 g/L, 24 h, 30 °C, pH 6.5, 125 rpm	-	25–200	73.0	[61]
Granular activated carbon	0.15 g/L, 24 h, 30 °C, pH 9	-	50–250	100.38	[74]
rGO/WO_3_	0.45 g/L, 80 min, 35 °C, pH 4, 200 rpm	48.823	10–50	73.05	This work

## Data Availability

The data presented in this study are available upon request from the corresponding author.

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
