# Peer review of "rGO-WO3 Heterostructure: Synthesis, Characterization and Utilization as an Efficient Adsorbent for the Removal of Fluoroquinolone Antibiotic Levofloxacin in an Aqueous Phase"

_molecules, 2022, doi:10.3390/molecules27206956_

Round 1

Reviewer 1 Report

1. Please give the full name where the first abbreviation appears in the abstract.

2. The first paragraph of introduction is too long. It is recommended to divide into several paragraphs.

3. Sustainable strategies for the treatment of wastewater and approcahes for the removal of fluoroquinolones must introduced comprehensively.

4. Why rGO-WO3 heterostructure is selected for LVX adsorption? Please explain from the relationship between structure and functions.

5. In line 221, the prepared rGO-WO3 heterostructure possessed the specific surface area 48.823 m2/g. As we know, the average specific surface area of graphene is 2630 m2/g. Why does the specific surface area decrease significantly, and does it have a great impact on the adsorption efficiency?

6. The fluorescence characterization of rGO-WO3 is redundant.

7. In addition to pseudo-second-order kinetic model, intraparticle diffusion model and pseudo-first-order model also need to be discussed and compared.

8. Please compare the adsorption paramaters and maximum adsorption capacity with other LVX adsorbents.

9. An important indicator for evaluating the adsorbnet is its application in actual samples. Please supplement the adsorption of LVX in actual samples.

10. The font size in figure captions should be uniform.

11. Error bars are needed in figure 9-14.

12. The format of the 3 tables should be uinform,  three-line tables are recommended.

13. There are to many figures in the manuscript, some pictures are suggested to be placed in supportion information.

Reviewer 2 Report

Comments to authors

The authors presented a manuscript titled “rGO-WO3 heterostructure: Synthesis, characterization and utilization as an efficient adsorbent for the removal of fluoroquinolone antibiotic levofloxacin in aqueous phase”. Impressively, a full range of characterization was done. Overall, the paper is scholarly. However, several key issues should be well addressed before the paper is considered for publication.

[1] Line 69-70 reads as thus “As graphene oxide (GO) exhibits porous structure functionalized with oxygen moieties which could provide base for the uniform loading of WO3 on its surface”. The authors have however prepared reduced graphene oxide that has less oxygen moieties than graphene oxide. This is contradictory and the authors need to address the contradiction.

[2] On the synthesis of rGO-WO3 heterostructure in 2.2 (line 94-104), graphene oxide (GO) was never reduced to reduced graphene oxide (rGO), but the final heterostructure was labelled as rGO-WO3. This is another contradiction that the authors need to address.

[3] The scale bar on the TEM and SEM images in Fig. 1-3 which we can use to try and measure the size of the nanostructures have been covered by the large nanometre numbers displayed on the images. The authors should either make the numbers smaller or move then far from the scale bar to enable readers to have evidence of their prepared nanostructures.

[4] Different font/font sizes were used in labelling figure 3. The authors should address this.

[5] The authors reported on line 188 “No apparent peak corresponded to GO was found due to its less amount”. It seems the authors understand that what they prepared is GO instead of rGO, but then they reported the final heterostructure as rGO-WO3 instead of GO-WO3 which is very confusing. Can the authors address this. Since they don’t have the XRD of their anticipated rGO, it becomes difficult to see if they managed to reduce GO to rGO. This requires further clarification.

[6] The O1s XPS data suggests that GO was formed instead of rGO looking at the very high presence of OH groups on the surface of the material. GO exhibits a lot of surface OH groups, as opposed to rGO that shows less OH groups and instead have more oxygen vacancies.

[7] Figure legends should be fixed in all the figures. Either different fonts or font sizes were used.

[8] The results of Pseudo-first order were not reported anywhere. How do we conclude that the equilibrium results fit Pseudo-second order without comparing the results to Pseudo-first order.

[9] Table 2 needs to be fixed.

[10] Why is it that only two-parameter isotherms were studied? What if the data fit with a 3/4/5 parameter isotherms? The paper below can assist you.

Ayawei, N., Ebelegi, A. N., & Wankasi, D. (2017). Modelling and interpretation of adsorption isotherms. Journal of chemistry, 2017.

[11] The long-term stability of the heterostructure was not studied. After how many adsorption-desorption experiments does the material starts to lose its performance capabilities and removal efficiency?

[12] The heterostructure was not applied to any real environmental samples, but the conclusion says, “the present work highlighted the significance of graphene derivatives based heterostructures in the field of environmental remediation”. This application of this statement on this work is insignificant as none of the work on environmental samples was carried out. Even if the authors say the materials works in synthetic samples, it has not been proven to work in complex matrices, which leaves many loopholes in this work.

Reviewer 3 Report

1.     Pls provide the SEM for the as-synthesized samples

2.     Make a labeling on the IR data

3. Is it possible for the significant role of the surface wettability of the materials to the high adsorption capability? as published in ACS Sustainable Chemistry & Engineering, 2016, 4 (12): 7013-7022. 4. Since pharmaceuticals and personal care products have huge production with extensive usage as well as frequently detected in surface water and ground water. How can the adsorbent be put into the underground water? How to control costs?

5. Some related refs could be compared and cited, such as Inorg. Chem. Commum., 2021, 130,108756, Cryst. Growth Des.2018, 18, 7114-7121, Dalton Trans. 2016,45, 15492-15499, Chem. Asian J. 2019, 14, 3694 –3701, Mater. Today. Commum., 2022,31,103514, CrystEngComm, 2021, 23, 8043–8052

6. Source and purity of all chemicals used should be specified in the experimental section. 7. The content of rGO in rGO- WO3 should be carried out in this manuscript and the relation illustration should be added in this section. 8. The structural stability should be confirmed by the morphology of rGO- WO3 samples after multiple adsorption tests. 9.The manuscript contains spelling/grammatical errors. So, the language should be polished thoroughly.

Round 2

Reviewer 1 Report

The suggestions put forward have been revised. The manuscript could be suitable for publication. 

Reviewer 2 Report

The authors addressed all the comments given to them to my satisfaction. The manuscript can be accepted for publication at its current state.

Reviewer 3 Report

It can be accepted based on the current revisions.